# Comparison of HIV-1 A6 dispersal dynamics in Poland before and after the war in Ukraine

**Karol Serwin[1]\*, Kaja Mielczak[1], Anna Urbańska[1], Bogusz Aksak-Wąs[1],
Malwina Karasińska-Cieślak[1], Piotr Ząbek[2], Ewa Siwak[3], Iwona Cielniak[4],
Elżbieta Jabłonowska[5], Paweł Jakubowski[6], Błażej Rozpłochowski[7],
Aleksandra Szymczak[8], Bartosz Szetela[8], Anna Kalinowska-Nowak[9],
Monika Bociąga-Jasik[9], Elżbieta Mularska[10], Adam Witor[10], Anita Olczak[11],
Władysław Łojewski[12], Maria Hlebowicz[13], Miłosz Parczewski[1]\***

**1** Department of Infectious, Tropical Diseases and Acquired Immunodeficiency, Pomeranian Medical University in Szczecin, Szczecin, Poland, **2** Molecular Diagnostics Laboratory, Hospital for Infectious Diseases, Warsaw, Poland, **3** Department of Infectious and Tropical Diseases and Hepatology, Medical University of Warsaw, Warsaw, Poland, **4** Faculty of Medical Science, Collegium Medicum Cardinal Stefan Wyszynski University in Warsaw, Warsaw, Poland, **5** Department of Infectious Diseases and Hepatology, Medical University of Łódź, Łódź, Poland, **6** Infectious Diseases, Pomeranian Hospitals, Gdańsk, Poland, **7** Department of Infectious Diseases, Hepatology and Acquired Immunodeficiencies, Karol Marcinkowski University of Medical Sciences, Poznań, Poland, **8** Department of Infectious Diseases, Liver Disease and Acquired Immune Deficiencies, Wroclaw Medical University, Wrocław, Poland, **9** Department of Infectious and Tropical Diseases, Jagiellonian University Medical College, Kraków, Poland, **10** Department of Infectious Diseases, Regional Hospital Chorzów, Chorzów, Poland, **11** Department of Infectious Diseases and Hepatology, Faculty of Medicine, Nicolaus Copernicus University Ludwik Rydygier Collegium, Bydgoszcz, Poland, **12** Department of Infectious Diseases, Regional Hospital in Zielona Góra, Zielona Góra, Poland, **13** Department of Family Medicine and Infectious Diseases, University of Warma and Mazury in Olsztyn, Olsztyn, Poland

\* karol.serwin@pum.edu.pl (KS); mparczewski@yahoo.co.uk (MP)

## Abstract

The war-related migrations from Ukraine to Poland have resulted in an increased prevalence of individuals diagnosed with human immunodeficiency virus type 1 (HIV-1) A6 variant. We examined the impact of the influx of people living with HIV (PLWH) displaced from Ukraine on the emergence of transmission events and evolving patterns in the A6 epidemic in Poland. We created a dataset of 13,696 unique HIV-1 *pol* gene fragments of sub-subtype A6 including 1,889 sequences from Poland. To evaluate the import of distinct clusters and estimate dispersal dynamics, we performed time calibration of the maximum-likelihood phylogenetic trees and phylogeographic inferences using the software package BEAST with discrete and continuous diffusion models. Our results indicated that A6 infections among males predominated within the domestic population (76.1%, n = 1,437), primarily within large clusters. Among Ukrainian migrants, 69.5% of the cases occurred as singletons or dyads (n = 473; p < 0.0001) with a balanced male-to-female ratio of 1.1. Since the war, the contribution of HIV-acquired individuals born in Ukraine to the virus circulation in Poland has increased to 30.2%, with an additional 334 distinct A6 introductions, inferred as internal nodes and descendant clusters that likely entered Poland from other countries.

**Data availability statement:** Patient confidentiality is paramount, especially when dealing with sensitive genetic data. While submitting real-life clusters to public databases provides an invaluable resource for research, it also carries a potential risk of breaching confidentiality. We have taken a conservative approach to mitigate this, adhering to the established protocols to protect individual privacy. As a result, only a random 10% sample of lineage A6 sequences has been submitted to GenBank. Sequence accession numbers were collected in Table I in S1 Text. The files required to re-run the phylogeographic analyses and the BEAST XML files are available at: https://github.com/Samer-Julo/HIV1-A6-in-Poland-circulation

**Funding:** This work was supported by the Medical Research Agency (Grant No. 2024/ABM/03/KPO/KPOD.07.07-IW.07-0042/24-00) under the framework of the National Recovery and Resilience Plan (Component D: Efficiency, accessibility, and quality of the healthcare system [Investment D3.1.1: Comprehensive development of research in the field of medical and health sciences]) granted to M.P. The funders had no role in the study design, data collection and analysis, the decision to publish, or the preparation of the manuscript.

**Competing interests:** The authors have declared that no competing interests exist.

These migration events were concentrated in the central regions with a higher HIV prevalence. After the war outbreak in 2022, the number and complexity of A6 transmission chains in Poland expanded, driven by male-dominated domestic clusters and war-related migration. Understanding the existence of two distinct transmission dynamics is critical for designing targeted public health interventions. Halting national sub-subtype A6 circulation requires a combined approach that harmonizes the existing strategy focused on the men who have sex with men population with enhanced efforts to link migrants to care.

## Author summary

The HIV-1 A6 sub-subtype is among the fastest expanding HIV epidemics in the World Health Organization's European Region, with a marked prevalence in Russia and Ukraine. In the last three years, vast population movements related to ongoing conflict in Ukraine have contributed to the increasing spread of the A6 variant across Europe, including Poland. Observed shifts present critical public health challenges, including the evolution of transmission and concerns regarding resistance to long-acting injectable therapies. The number of people born in Ukraine and receiving HIV care in Poland who were diagnosed with the A6 sub-subtype has increased significantly, especially among women. Since the war began in 2022, new cases have boosted the HIV epidemic in Poland. New transmissions of the A6 lineage have been found in the densely populated and urbanized areas of central and western Poland. Almost one-third of relative migration links were associated with people living with HIV (PLWH), born in Ukraine. Displaced individuals with HIV who arrived in Poland tend to move frequently, which has contributed to the circulation of the virus since 2022. To our knowledge, this study is the first to investigate the impact of mass population displacement from Ukraine on the molecular evolution of the HIV epidemic in the sheltered country, utilizing an extensive dataset of sequences from a national cohort of Ukrainian PLWH who have been displaced due to the war.

## Introduction

One of the most rapidly expanding human immunodeficiency virus type 1 (HIV-1) epidemics is the A6 sub-subtype, which predominantly affects regions that were formerly part of the Soviet Union [1]. Sub-subtype A6 represents one of seven phylogenetically distinct sub-lineages within subtype A of HIV-1 group M [2]. The A6 lineage originated from strains introduced into Odessa, Ukraine, and subsequently expanded as the $A_{FSU}$ lineage (Former Soviet Union A lineage) among people who inject drugs (PWID) since 1993 [3]. In the World Health Organization's (WHO) European Region, 72% of new HIV-1 infections in 2022 originated from the Eastern part of the continent, with the highest incidences reported in Russia and Ukraine (38.4 and 29.8 per 100,000 inhabitants,

respectively) [4]. In Poland, the prevalence of HIV-1 A6 has been increasing for over a decade, with a notable surge occurring after 2022, primarily attributed to the influx of people living with HIV (PLWH) displaced from Ukraine [5]. From the onset of the war in Ukraine in late February 2022 to mid-May 2025, 994,180 displaced people (primarily women and children) were registered under the national refugee protection schemes in Poland [6]. During this period, Poland recorded a substantial number of new HIV diagnoses and provided care to over 3,500 Ukrainian individuals living with HIV [7]. Migration after 2022 has directly impacted therapeutic complexity in Poland by expanding the A6 sub-subtype, which represents a genetically distinct lineage of group M HIV-1 than historically dominant subtype B [8]. Observed shift has contributed to a relative increase in heterosexually acquired HIV cases, along with greater inclusion of women, individuals with hepatitis C or tuberculosis co-infection, and late HIV diagnoses [5]. Additional challenges faced by individuals arriving in Poland include healthcare-seeking behaviours and expectations, as well as language barriers, which may affect their engagement in care and adherence to treatment [9]. Furthermore, the co-circulation of A6 and B subtypes increases the likelihood of the emergence of novel A6/B recombinant forms, which now account for 1.8% of all sequenced HIV-1 *pol* gene samples in Poland [10].

Molecular HIV epidemiology in other regions of Europe has also been affected by migration patterns, with notable influence of mobility from Ukraine, as evidenced by the recent surge in HIV-1 A6 infections in Sweden [11]. Previous discrete phylogeographic inference revealed a complex history of viral migration between Ukrainian and Polish regions before 2022 [12]. The migration-driven changes in HIV subtype distribution, transmission dynamics, and therapeutic complexity are observed not only in Poland but in all European countries with large populations of war-related migrants. A review of strategies for treatment and public health surveillance across Europe is evolving, with increasing attention to migration-related background and the demographic structure of displaced populations [13].

The HIV-1 A6 lineage has been identified as a potential risk factor for virological failure to the long-acting injectable HIV treatment containing cabotegravir/rilpivirine (CAB/RPV) [14,15]. This association is supported by recent evidence showing that the L74I polymorphism, a signature mutation of A6 sub-subtype, may contribute to cabotegravir resistance selection when present alongside integrase resistance mutations [16,17]. Furthermore, infection with the A6 sub-subtype, in combination with baseline non-nucleoside reverse transcriptase inhibitor (NNRTI) resistance and higher body mass index (BMI), was determined as an independent predictor of virologic failure in clinical settings [14]. To date, resistance-associated mutations to CAB/RPV remain uncommon among naive and treatment-experienced individuals with the A6 sub-subtype born in Poland [18]. Notably, the A6 variant is the most prevalent lineage among migrant populations diagnosed in Poland. Further studies are required to determine the potential impact of these cases on the efficacy of long-acting injectable HIV medications [19].

Large migration flows may expand and alter the previously identified transmission chains of the A6 sub-subtype [12,20]. We anticipated further introductions since the onset of the war in 2022, which could significantly increase the burden of HIV in Poland. In this study, we analyzed the spatiotemporal distribution of the HIV-1 A6 sub-subtype in Poland, with particular attention paid to PLWH displaced from Ukraine and later diagnosed in Poland after the outbreak of the full-scale war.

## Methods

### Ethics statement

The study protocol was approved by the Bioethical Committee of the Pomeranian Medical University, Szczecin, Poland (approval numbers: KB-0012/26/17 and KB-0012/08/12). All patients provided written informed consent for sample collection and clinical data processing to conduct this study. All samples were de-identified to maintain the participant's anonymity. The study was conducted in accordance with the principles of the Declaration of Helsinki.

### Collation of a sequence dataset

We assembled an extensive dataset of HIV-1 sub-subtype A6 *pol* gene fragments (positions 2253–3554 of the HXB2 reference genome K03455), including genes encoding protease and part of the reverse transcriptase. The inclusion criteria for

the sequences are listed in the S1 Text. The final dataset contained 13,696 sequences: 1,889 from Poland and 11,807 from the background samples (Table A in S1 Text). Polish sequences were obtained from treatment- naïve and treated individuals between 1997 and November 2023 and represent all genotyped A6 samples from national HIV/AIDS centers. Publicly available HIV-1 A6 *pol* sequences from 48 countries were obtained from the Los Alamos National Laboratory (LANL) HIV database [2]. From the final alignment, duplicate entries were removed to avoid potential redundancy and to retain only the oldest sequence per individual. Finally, the alignment was edited to exclude 43 codon positions associated with drug resistance [21].

## Phylogenetic inference with temporal scaling

We applied previously established analytical workflows as described by Cuypers [22] et al. and Dellicour et al. [23]. An exhaustive background was included to analyze and identify Polish clusters within the global context of HIV-1 A6 phylogenetic diversity. The time-scaled phylogeny was inferred through a three-step process. First, we calculated the maximum likelihood (ML) phylogeny using IQ-TREE 2.3.6 under the general time reversible (GTR) model with four gamma-category sites [24]. We then assessed the temporal signal of the dataset using TempEst 1.5.3 [25], and the ML tree was inspected for outlier sequences (Fig A in S1 Text). Finally, the resulting ML tree was time-calibrated using TreeTime 0.11.1 [26] with an evolutionary rate of 0.00323 substitutions per site per year (s/s/y), as reported by Nasir et al. This rate was derived from 426 A6 variant sequences sampled from FSU countries between 1997 and 2016, and it exceeded the global median for subtype A *pol* sequences [27].

## Tracing of distinct introduction events in Poland

A preliminary phylogeographic analysis was conducted to identify internal nodes and descendant clusters potentially corresponding to distinct introductions into Poland, followed by subsequent circulation of the A6 lineage across the country [23]. We defined a cluster as a local transmission chain in Poland, initiated by a distinct introduction event, when the most probable location assigned to an internal node in the time-scaled phylogenetic tree was "Poland," and the location of its parent node was "non-Poland" [28]. These estimates were based on the topology of an approximate time-scaled phylogenetic tree obtained in the previous step. Bayesian inference via the Markov chain Monte Carlo (MCMC) technique was run for one million iterations with sampling every 1000 steps. Adequate mixing and convergence were confirmed by ensuring that effective sample sizes (ESS) exceeded 200 for all relevant parameters. The resulting maximum clade credibility (MCC) tree was used to characterize the phylogenetic clusters corresponding to independent introduction events in Poland. An introduction event was recorded when a node was assigned to "Poland" and its parent node in the tree to "non-Poland". We categorized the distribution of sequences into three cluster sizes: (i) large clusters ($n > 14$), (ii) networks ($2 < n < 14$), and (iii) singletons ($n = 1$) and dyads ($n = 2$)(together referred to as S&D); because clusters of different sequence proportions exhibit assorted dynamics during the HIV epidemic [29]. Large clusters reflect long-standing domestic transmission chains, whereas networks capture ongoing but smaller-scale transmission with a more recent common ancestor. Singletons and dyads represent new introductions, indicating emerging outbreaks or importation of genetically distinct viral strains.

## Phylogeographic reconstruction of the domestic A6 lineage circulation

We conducted discrete and continuous phylogeographic estimations of the clusters identified in the previous step, each containing at least three sequences circulating within the country (i.e., networks and large clusters). The resulting clusters represent local transmission chains initiated by distinct entry points into the study area. We applied the Bayesian stochastic search variable selection (BSSVS) method using the BEAST 1.10 package to identify the most strongly supported lineage migration events between regions [28]. For viral movement between traits, we denoted the origin region as "from" and the recipient region as "to". The same approach was used for migration events between different population origins (i.e., local or migrant). The MCMC chains were run until all parameters achieved ESS values $> 200$,to ensure adequate

mixing. To account for sampling bias due to the relative abundance of samples by region, we incorporated the adjusted Bayes factor (BFadj), which evaluates both the expected and observed inclusion frequencies under the BSSVS and requires two separate analyses [30]. Linkages with BFadj support of ≥3 were considered significant [31]. For continuous analysis, we utilize BEAST 1.10 with the relaxed random walk (RRW) diffusion model [32]. As our data included only the first administrative level coordinates, we assigned geographic locations to each sample by randomly selecting a point within its region of origin. Lastly, the R package "Seraphim" was employed to extract spatiotemporal information integrated within posterior trees and visually present the continuous phylogeographic simulations [33].

## Results

### Sequence data overview

For our analysis, we obtained 1,219 (64.5%) sequences originating from individuals of Polish nationality (locals, born in Poland), while 670 (35.5%) were from Ukrainian migrants (born in Ukraine). A detailed description of the sample breakdown is provided in Fig B and Table B in S1 Text. Sequences from men accounted for more than three-quarters of the cases (n = 1,438, 76.1%) in the studied cohort, with men who have sex with men (MSM) being the most commonly disclosed risk factor (n = 387, 20.5%)(Fig 1). Until 2021, sequences from men were predominant in both nationality groups: PLWH born in Poland (Fig C in S1 Text) and those born in Ukraine (Fig D in S1 Text). However, in 2022 and 2023, a higher number of sequences was recorded among women in the group of migrants from Ukraine.

### Determining distinct introduction events and phylogenetic clusters of the A6 variant circulating in Poland

We identified clusters and singletons representing samples with separate entry points into Poland. A total of 670 lineage introductions (95% highest posterior density [HPD] interval = 663–678) were identified. The highest number of

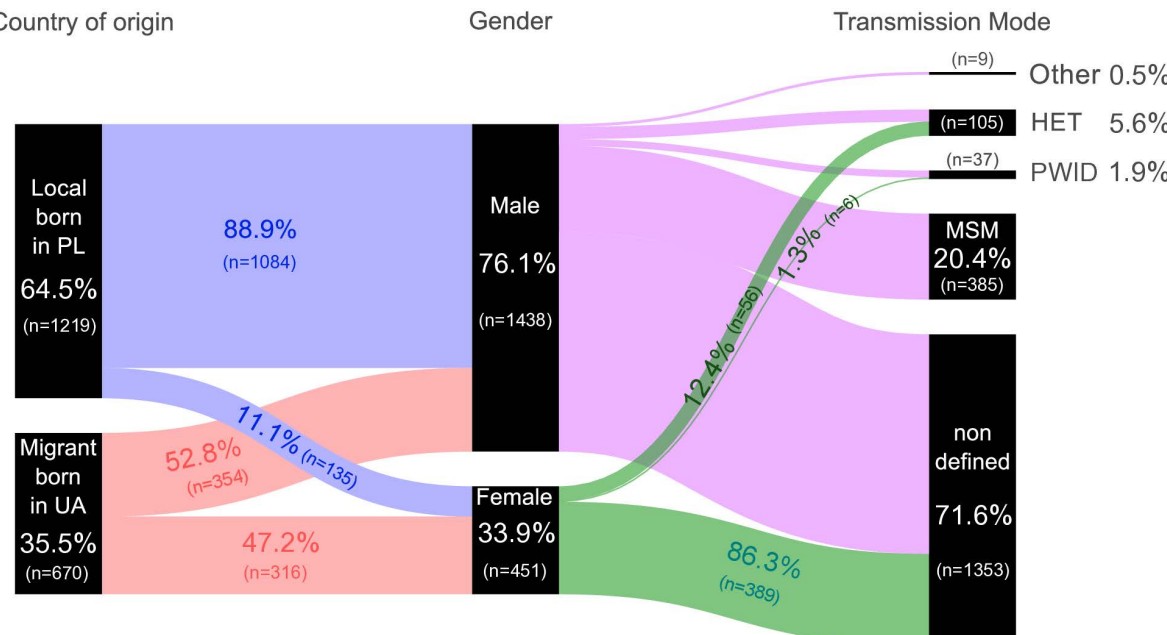

**Fig 1. Alluvial diagram shows from left to right the HIV-1 A6 sequence distribution among people linked to care in Poland by country of origin, gender, and risk factor; PL – Poland/Polish; UA – Ukraine/Ukrainian; MSM – men who have sex with men; HET – heterosexual individuals; PWID – people who inject drugs.**

introduções was recorded in 2022 (n = 177, 26.4%; Fig E.1 in S1 Text). Preliminary discrete phylogeographic analysis identified six large clusters (n = 855 sequences, 45.3%), 74 networks (n = 352 sequences, 18.6%), and 590 singletons or dyads (n = 682 sequences, 36.1%). In 2022, a substantial increase in the incidence of HIV A6 was observed across all specified cluster sizes (Fig 2A). Large, long-lasting clusters remained consistently predominant in the Polish A6 sub-subtype epidemic, with individuals born in Poland accounting for 95.5% (n = 817) of the cases. However, a significant

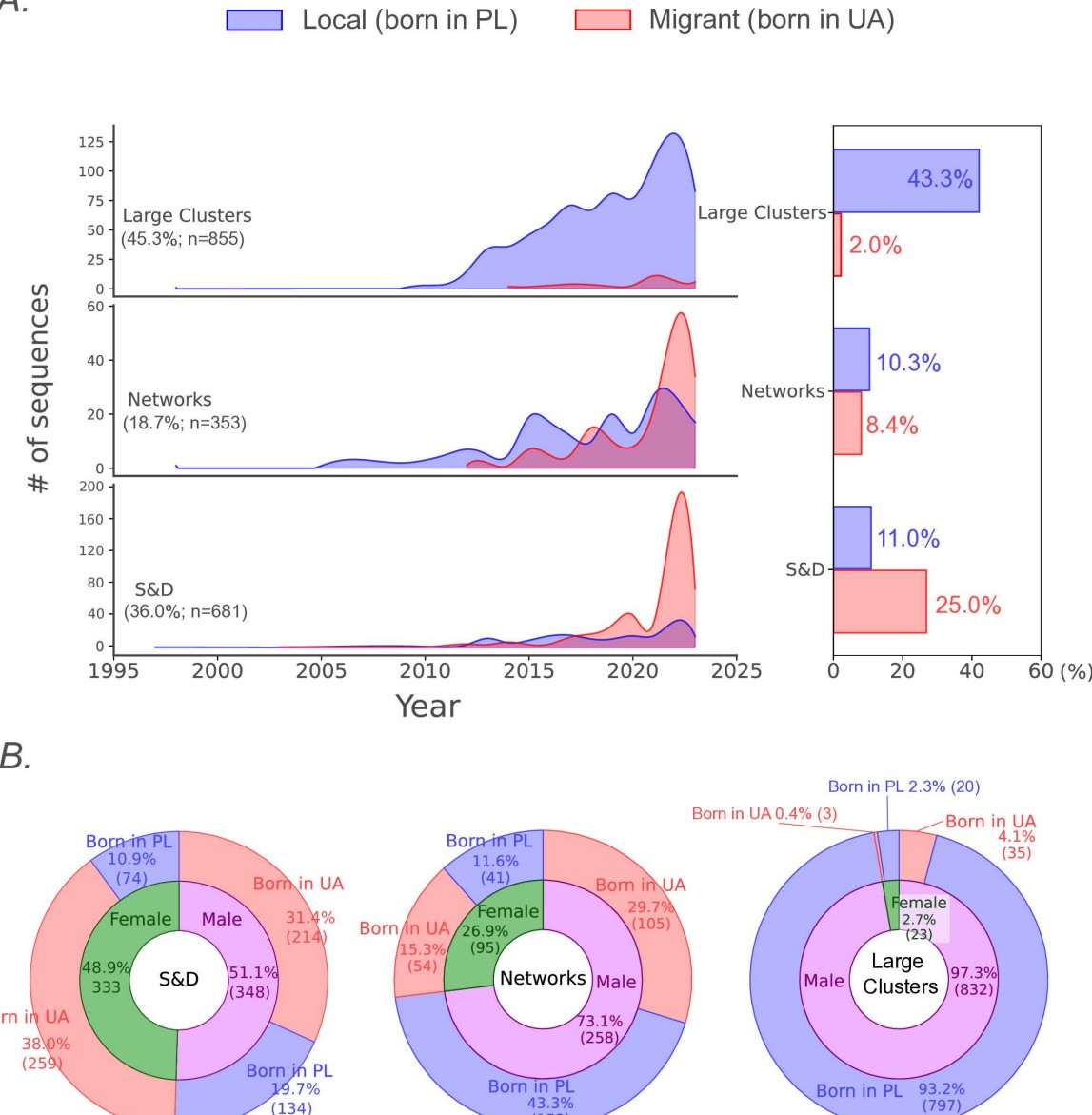

**Fig 2. Characteristics of HIV-1 A6 sequence distribution among identified cluster sizes.** (A) Sampling density by cluster size and country of origin. Cluster categories are defined as Large Clusters (n > 14), Networks (3 ≤ n ≤ 13), Singletons (n = 1), and Dyads (n = 2) – S&D; (B) Donut charts showing gender and country of birth distribution for various cluster sizes; HET – heterosexual individuals; MSM - men who have sex with men; PWID - people who inject drugs.

shift was observed in the last 2 years, with sequences from PLWH born in Ukraine becoming predominant within the networks. A considerable number of cases were observed among Ukrainian migrants, particularly singletons and dyads (69.4%, n = 473; p < 0.0001), indicating the most recent introductions of the A6 lineage into Poland. Notably, the proportion of females was significantly higher among Ukrainian migrants than the Polish-born population (47.1%, n = 316 vs. 11.1%, n = 135; p < 0.0001) (Fig 2B).

**Phylogeographic Inferences of HIV-1 Sub-subtype A6 in Poland**

First, we used discrete phylogeography to uncover distinct dispersal patterns of the A6 sub-subtype across various regions, nationality groups, and time frames.  Overall, the Masovian region acted as the country's central spot, accounting for 60.0% as a source and 59.6% as a sink destination for migration events. Other regions had a smaller share of the dispersal of the A6 lineage, including Lesser Poland (11.7% source, 8.9% sink) and the West Pomeranian (7.3% source, 7.7% sink)(Fig 3A). A systematic analysis of the transmission dynamics across diverse historical periods and geographical regions is provided in the in S1 Text and Table C in S1 Text.

As the geographical movement and distribution patterns of the two population groups (local, i.e., born in Poland; migrants, i.e., born in Ukraine) were inferred simultaneously, we examined the relationship between migration locations and these national groups. Overall, migration links between PLWH born in Poland were predominant (from local-to-local transmission type, Fig 3A), representing an average of 88.3% of migration events (with the highest rate of 97.1% recorded between 2007 and 2012; Table C in S1 Text). A relatively high number (6.6%) of migration events was observed between 2017 and 2022 for migrant-to-migrant connections, with 3.6% of these links located within the Masovian region (Fig 3B). We estimated that the greatest increase in the relative contribution of migration events involving PLWH born in Ukraine (either between migrants or between migrant and local modes) occurred after 2022, accounting for 30.2%, of these events. After 2022, the most substantial increase was observed in migration events between PLWH born in Ukraine (migrant to migrant mode), representing 21.1% of the events, particularly in the West Pomeranian, Lodz and Greater Poland regions (Fig 3C). Additionally, during this period, we observed an increased number of links (7.5%) between PLWH born in Poland and those born in Ukraine (local-to-migrant linkages).

Continuous phylogeographic reconstructions were broadly consistent with the discrete analyses and showed similar dispersal patterns. Samples from the Northwestern (West Pomeranian, Pomeranian) and Southern regions (Lesser Poland, Silesian, Lower Silesian) regions were mainly connected to the geographic center of the country, around the Masovian and Lodz areas (Fig 4A–C; Tables D and E in S1 Text). Our analysis indicated that migration events were primarily concentrated within regions characterized by relatively higher local circulation during each examined period (Fig E.2 in S1 Text). Both discrete and continuous phylogeographic calculations revealed that the majority of migration events occurred between 2017 and the beginning of 2022 (on average, 43.6%), however the highest yearly density of events was recorded in 2022 (11.5%) (Fig E.3 in S1 Text).

**Singletons and dyads boost dynamics of the HIV-1 A6 epidemic in Poland after onset of the war in Ukraine**

Phylogeographic analysis provided insights into movement between locations, focusing on clusters with at least three sequences. In our study, these were represented by large clusters (n = 6, 0.9%) and networks (n = 74, 11.0%), as described in the previous section. However, many sequences were classified as singletons (n = 498, 74.3%) or dyads (n = 92, 13.8%)(Fig 2A). To gain insight into the dispersal patterns of the two population groups (PLWH born in Poland or Ukraine), we analyzed the distribution of singletons and dyads by country of origin, specifying the beginning of 2022 as a turning point associated with the conflict in Ukraine. Before 2022, most sequences were collected from the Central (37.3% from Masovian and 9.3% from Lodz) and Northwestern (15.2% from West Pomeranian and 6.2% from Pomeranian) regions (Fig 5A). After 2022, the interior regions continued to contribute a notable share of new infections (24.2% from Masovian and 8.6% from Lodz), while a shift towards Western areas was observed, with the West Pomeranian (16.8%),

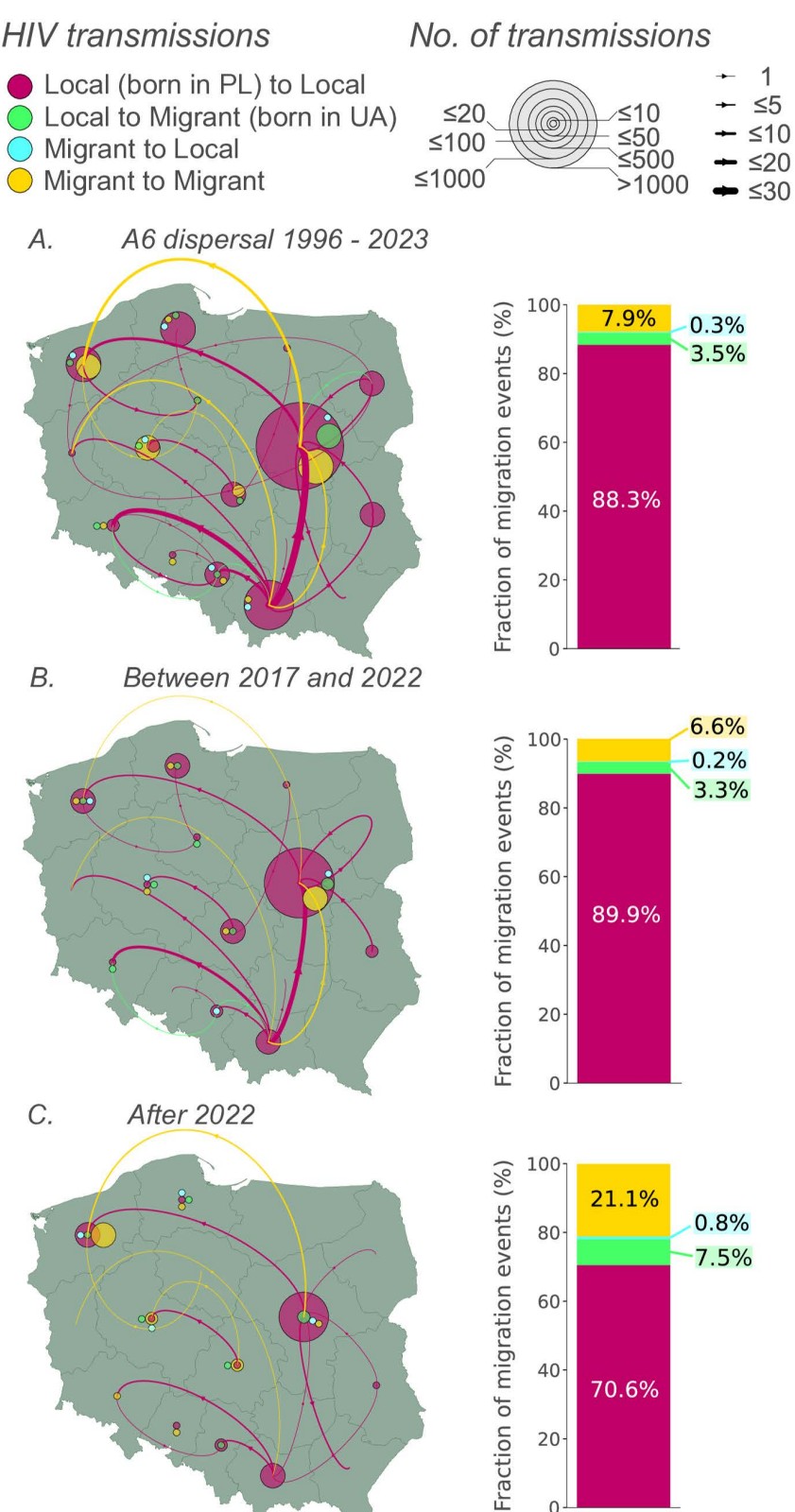

**Fig 3. Discrete phylogeographic reconstruction of A6 lineage migration events between regions and national groups (PLWH diagnosed in Poland born in Poland or Ukraine) over time with a bar graph depicting the relative contribution of population groups to the spread of the A6**

**sub-subtype: (A) all migration events from 1996 to 2023, (B) between 2017 and 2022, (C) after 2022.** Arrow thickness corresponds to the average number of inferred migration events between regions, while transparent circle sizes represent lineage dispersal events inferred within regions. Calculations were averaged over posterior trees sampled from each posterior distribution. Inward movements between specific groups (PLWH born in PL - Locals and PLWH born in UA - Migrants) are depicted in the same color. Only migration events with an adjusted Bayes factor support ≥3 are reported. Abbreviations: HIV, human immunodeficiency virus; PL, Poland/Polish; UA, Ukraine/Ukrainian. Detailed information is found in Table C in S1 Text. Base layer source: GADM database of Global Administrative Areas, version 4.0, available at https://gadm.org.

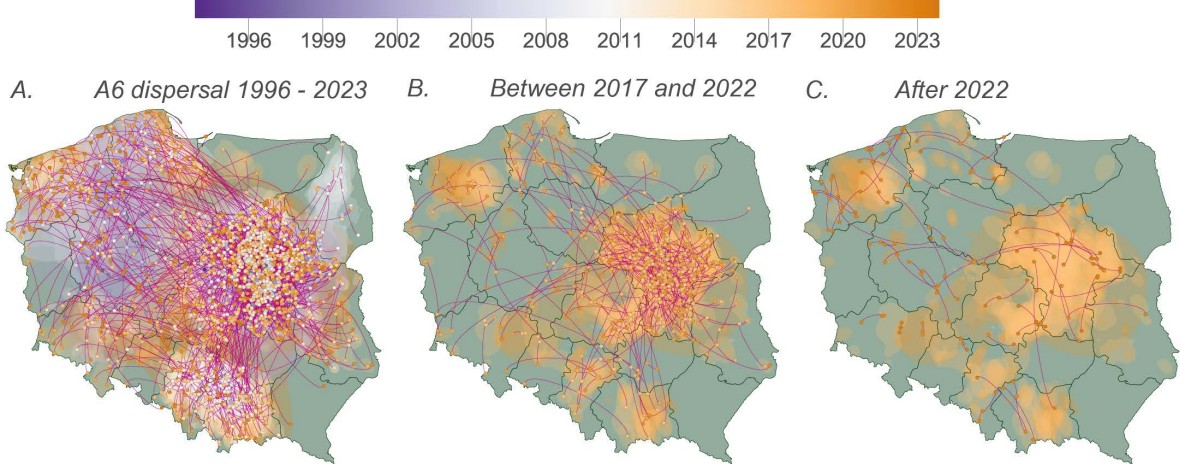

*Timing of transmissions*

**Fig 4. Continuous phylogeographic reconstruction of migration events between regions in Poland over time: (A) all migration events from 1996 to 2023, (B) between 2017 and 2022, (C) after 2022.** We depict the maximum clade credibility (MCC) tree and the 80% highest posterior density (HPD) regions to illustrate the uncertainty associated with the Bayesian inference. The nodes of the MCC tree are color-coded according to their temporal occurrence, and the 80% HPD regions were calculated for successive time intervals and overlaid using a consistent color scale to reflect time. Base layer source: GADM database of Global Administrative Areas, version 4.0, available at https://gadm.org.

Greater Poland (15.6%), and Lower Silesian (12.4%) being more frequently represented (Fig 5B). The vast majority of singletons originating from migrants (born in Ukraine) were registered after 2022 (86.6%, n = 219) compared to earlier years (60.8%, n = 149; p < 0.0001)(Fig 5C and Table F in S1 Text). Before the war outbreak in Ukraine, the most common type of dyad linkage was observed among local cases (44.2%, n = 19; p < 0.05), primarily emerging within regions (Table G in S1 Text). After 2022, the majority of dyads were observed as connections among migrants born in Ukraine (62.8%, n = 27; p < 0.001), exhibiting intra- and cross-regional spread (Table H in S1 Text).

## Discussion

We reconstructed the migration events in the country involving PLWH born in Ukraine based on large clusters and networks. Before the war, migrants accounted for 7.9% of the national circulation of sub-subtype A6, which increased to 11.7% by 2022. War-displaced people have also expanded the geographical coverage of the A6 lineage transmission chain. Before 2022, 10.6% of interregional migration events involved PLWH born in Ukraine; after the outbreak of the conflict, this proportion increased to 15.7%. A similar observation can be made regarding the distribution of singletons and dyads originating from PLWH born in Ukraine. In just 2 years from early 2022, the number of singletons and dyads increased by 14.1% and 58.1%, respectively, compared to the overall (26-year) sampling period before the war. Notably, we found dyads involving cross-regional connections between migrants originating from Ukraine only after 2022.

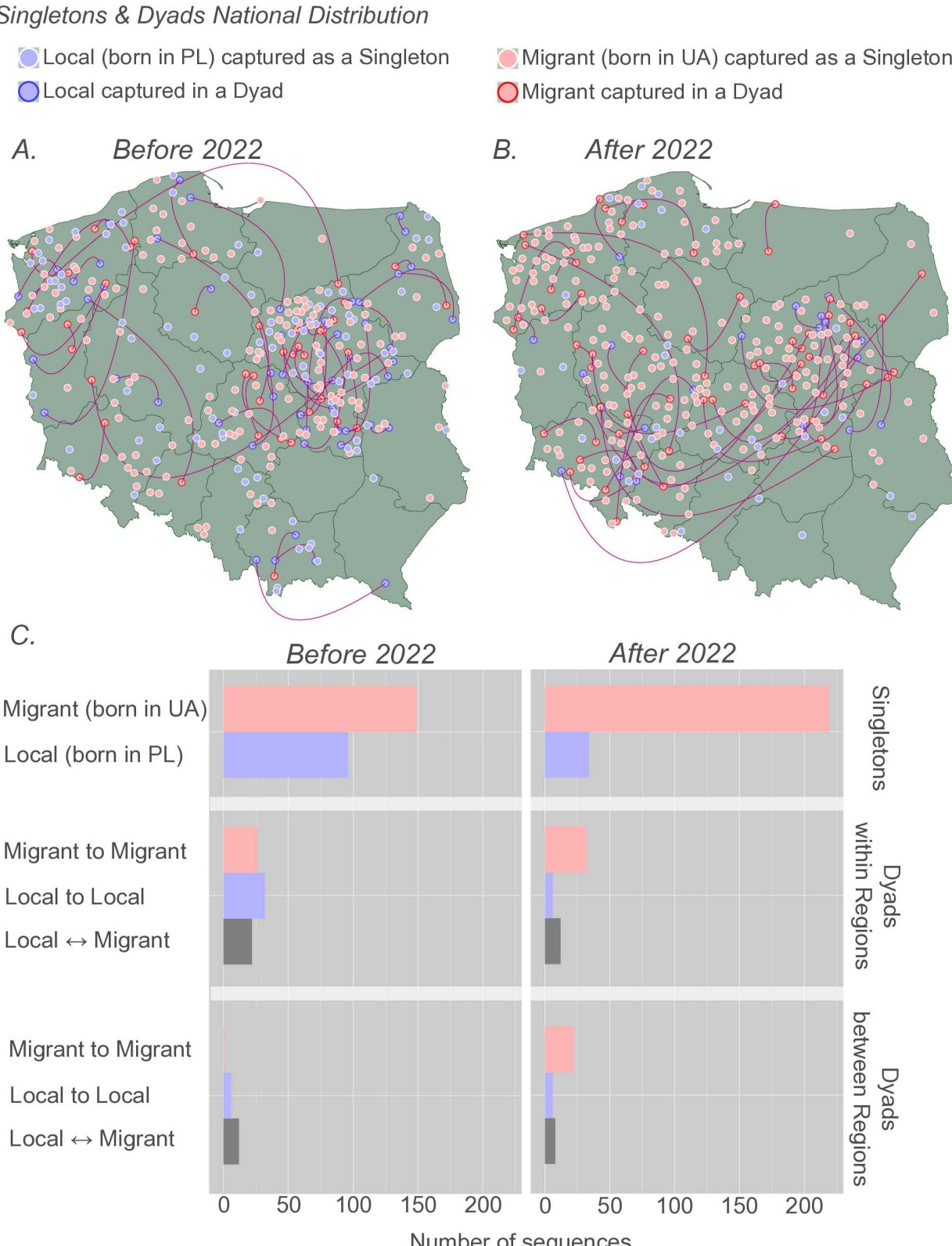

**Fig 5. Investigation of spatio-temporal distribution of singletons and dyads.** Phylogenetic clusters identified from the maximum clade credibility (MCC) tree, including n = 1 (singletons) and n = 2 (dyads) sequences, are randomly placed on the map, maintaining regional specificity. Points represent

individual sequences colored by origin: Local (born in Poland - blue) and Migrant (born in Ukraine - red). Lines represent phylogenetic branches between dyads. The points and lines in this figure do not represent migration events. Instead, they illustrate the spatial placement of singletons and dyads based on their sampling region. (A) Singletons and dyads before 2022; (B) singletons and dyads after 2022; (C) distribution of singletons and dyads across two population groups, including dyad linkage composition and intra- or inter-regional dispersal before and after 2022; Base layer source: GADM database of Global Administrative Areas, version 4.0, available at https://gadm.org.

Considering the above observations, PLWH who arrived in Poland after the outbreak of the war demonstrated high mobility and contributed to the expanding transmission chain since 2022.

A large numbers of females were found in networks, dyads and singletons. Minor clusters were either imported or developed, mainly in the near past, representing recent or ongoing HIV transmission. Females constitute the majority of migrants, especially those arriving after 2022. This trend was observed across the EU/EEA region, mainly due to crossing border regulations for Ukrainian men aged 18–59 years [34]. As a result, an increase in heterosexually acquired HIV cases is expected, along with the growing inclusion of women in transmission chains.

Our previous analyses, which focused exclusively on inter-regional transmissions before 2022, also identified Central Poland as the primary center of the A6 lineage circulation [12,35]. The transmission profile of the A6 variant involving PLWH born in Ukraine is broadly consistent with the observed flow of Ukrainian migrants across different regions of Poland. In 2021, 70.5% of Ukrainian citizens recorded in Poland were concentrated in six regions, with the highest numbers in Masovian (23.9%), Lesser Poland (15.5%), and Greater Poland (8.9%) [36]. In our phylogeographic analysis, Masovian and Lesser Poland served as central hubs for A6 dispersal. After 2022, changes in the distribution of Ukrainian migrants were observed: the largest increases in the number of Ukrainian citizens compared to 2021 (ranging 2.4 - 3.2 times) were noticed in West Pomerania, Greater Poland, and Lodz [36]. The movement of Ukrainian individuals had a particular impact on the dynamics of A6 transmission, after 2022, the substantial increase in migration events involving PLWH born in Ukraine was recorded in West Pomerania, Lodz, and Greater Poland, along with additional shifts in the distribution of singletons and dyads towards the Western regions.

Excluding infections among PLWH born in Poland, it is evident that after 2022, new transmissions occurred primarily in a migrant-to-migrant pattern; however, bidirectional networks between natives and migrants have also been reported. It is possible that the migrant-to-migrant links observed in Poland involved individuals who acquired HIV within local transmission chains while living in the same areas of Ukraine before moving to shelter destinations in Poland [12]. Although our dataset included all publicly available sequences from Ukraine, the limited sampling effort in Ukraine in recent years, primarily because of the current war, restricts the ability to infer all connections. Some singletons and dyads may reflect either isolated introductions or the early stages of local transmission chains that remain undetected due to incomplete sampling and could expand over time, particularly in the context of ongoing migration and changing epidemic dynamics. Consequently, any conclusions regarding potential cross-border transmission should be viewed with caution, given the current limitations and gaps in available sequence data. Moreover, the European Centre for Disease Prevention and Control (ECDC) estimates that an additional 10,000–30,000 PLWH are likely to move from Ukraine to the EU/EEA region [37]. Given that 27.6% of all Ukrainian war refugees have sought shelter in Poland [38], the estimated number of imported HIV cases is likely to range between 2,800 and 8,300, underscoring the continuing need for sufficient linkage to care and testing among war displaced individuals. The ongoing large-scale population displacement has imposed an increased burden on HIV care delivery and public health services, a pattern that has also been documented in Germany, the Czech Republic, and other countries adjacent to Ukraine [9].

An increasing number of PLWH resettled from Ukraine represent separate introductions of the virus into Poland, corresponding to different branches of the international transmission chain with various entry points into the country. We noted a substantial rise in virus introductions to Poland, with 336 cases before 2022 [35] and an additional 334 importations in the following 2 years. Monitoring A6 transmission dynamics is essential for treatment planning, because the presence

of specific polymorphisms, such as L74I and other integrase resistance mutations, may influence the effectiveness of antiretroviral therapies, including long-acting CAB/RPV, and contribute to the risk of virological failure [39], especially in regions like Eastern Europe, where HIV-1 A6 infections are prevalent and continue to increase. Therefore, subtyping and sequence analysis should be considered for all migrants with virological symptoms to assess drug resistance and to follow epidemiological changes. The inclusion of migration-related background in molecular surveillance at the population level may enhance national capacity to track HIV transmission, inform contact tracing efforts, and guide context-specific public health measures. Coordination of treatment and monitoring strategies across European health systems remains important for responding to the changing landscape of the HIV epidemic.

Considering the limitations of this study is essential. Primarily, specific locations may be underrepresented because of the varying sampling efforts employed across domestic HIV/AIDS centers. The samples were collected mainly by analyzing drug resistance at the time of care entry (baseline) or treatment failure. Moreover, data on Ukrainian migrants linked to care in Poland are likely to be underestimated, and transmission mode details are absent in many cases, precluding a dynamic analysis of risk groups. Nevertheless, the present study encompasses an extensive dataset of sequences from diverse cohorts and geographical regions within Poland. Subsequently, our analysis was limited to a fragment of the *pol* gene and breakpoints outside this region could not be detected. Other HIV-1 subtypes and recombinant forms are also present in Poland and contribute to increasing genetic diversity and epidemic complexity. Next, the phylogenetic tree and ancestral traits of the corresponding internal nodes were inferred separately, and the evolutionary rate used for time calibration was derived from a prior estimate, not directly calculated from the current dataset. Assuming the computational constraints of our large dataset, this approach represented a balance between conducting phylogeographic analyses on large-scale datasets and ensuring robust results; however, it may introduce assumptions that could influence the time-scale estimates [23]. To address the limitations of the sequence dataset, we adjusted the model sensitivity by applying biased sampling to the subpopulations. A filter based on the proportional representation of samples from each trait was implemented, which closely resembled the tip-date randomization test for temporal signals and considered the frequency of sampling by location [30].

## Conclusions

The number and structure of A6 transmission chains in Poland have increased since 2022. Importing of the virus through war-related migration further intensified the A6 epidemic in Poland. The expansion of migration events presents a vital challenge for the national health system, particularly because of resources are constrained by the need to support the large influx of individuals from abroad, placing additional pressure on clinical services, medical personnel, and broader public health infrastructure. HIV/AIDS centers, particularly in Central and Western Poland, need to adjust the appropriate healthcare services that have the potential to reduce transmission chains within the native MSM populations and include testing and prevention to control additional dissemination involving migrant individuals. Genotypic resistance testing should be considered for all migrants with virological symptoms to monitor epidemiological changes, particularly because the findings indicate multiple distinct A6 imports.

## Supporting information

**S1 Text.** Table A. Distribution of sequences per country in the background data set; Table B. The distribution of sequences per region in the A6 Polish data set; Table C. The detailed breakdown of HIV-1 A6 lineage migration events inferred between regions and nationality groups of PLWH in Poland; Table D. The detailed breakdown of HIV-1 A6 lineage migration events by nationality groups of PLWH diagnosed in Poland; Table E. A detailed breakdown of HIV-1 A6 lineage migration events inferred between regions in Poland; Table F. Distribution of singletons across regions before and after 2022; Table G. The phylogenetic branches between individuals captured in dyads across regions in Poland before 2022; Table H. The phylogenetic branches between individuals captured in dyads across regions in Poland after 2022; Table

I. GenBank accession numbers for a representative 10% subset of the polish HIV-1 A6 sequences. Fig A. Root-to-tip regression analysis performed using TempEst 1.5.3. Fig B. Sampling density of the A6 sub-subtype in Poland; Fig C. The annual distribution of HIV-1 A6 sequences sampled in Poland among PLWH born in Poland; Fig D. The annual distribution of HIV-1 A6 sequences sampled in Poland among PLWH born in Ukraine; Fig E. Characteristics of inferred HIV-1 A6 circulation.
(DOCX)

## Author contributions

**Conceptualization:** Karol Serwin, Miłosz Parczewski.

**Data curation:** Karol Serwin, Kaja Mielczak, Anna Urbańska, Piotr Ząbek.

**Formal analysis:** Karol Serwin, Kaja Mielczak, Bogusz Aksak-Wąs, Miłosz Parczewski.

**Funding acquisition:** Miłosz Parczewski.

**Investigation:** Karol Serwin, Miłosz Parczewski.

**Methodology:** Karol Serwin.

**Project administration:** Karol Serwin, Miłosz Parczewski.

**Resources:** Bogusz Aksak-Wąs, Malwina Karasińska-Cieślak, Piotr Ząbek, Ewa Siwak, Iwona Cielniak, Elżbieta Jabłonowska, Paweł Jakubowski, Błażej Rozpłochowski, Aleksandra Szymczak, Bartosz Szetela, Anna Kalinowska-Nowak, Monika Bociąga-Jasik, Elżbieta Mularska, Adam Witor, Anita Olczak, Władysław Łojewski, Maria Hlebowicz, Miłosz Parczewski.

**Software:** Karol Serwin.

**Supervision:** Miłosz Parczewski.

**Validation:** Karol Serwin, Kaja Mielczak, Anna Urbańska, Piotr Ząbek, Miłosz Parczewski.

**Visualization:** Karol Serwin.

**Writing – original draft:** Karol Serwin.

**Writing – review & editing:** Karol Serwin, Kaja Mielczak, Anna Urbańska, Bogusz Aksak-Wąs, Malwina Karasińska-Cieślak, Piotr Ząbek, Ewa Siwak, Iwona Cielniak, Elżbieta Jabłonowska, Paweł Jakubowski, Błażej Rozpłochowski, Aleksandra Szymczak, Bartosz Szetela, Anna Kalinowska-Nowak, Monika Bociąga-Jasik, Elżbieta Mularska, Adam Witor, Anita Olczak, Władysław Łojewski, Maria Hlebowicz, Miłosz Parczewski.

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
