## [Decision Letter · Decision Letter 0]

25 Apr 2025

PPATHOGENS-D-25-00854

Comparison of HIV-1 A6 Dispersal Dynamics in Poland Before and After the War in Ukraine

PLOS Pathogens

Dear Dr. Serwin,

Thank you for submitting your manuscript to PLOS Pathogens. After careful consideration, we feel that it has merit but does not fully meet PLOS Pathogens's publication criteria as it currently stands. Therefore, we invite you to submit a revised version of the manuscript that addresses the points raised during the review process especially those related to the phylogenetic analyses. We ask that you specifically address the issues raised regarding the time calibration of the phylogenetic tree, details on the use of BEAST and phylogenetic networks, and the possibility of HIV-1 subtype recombination. 

Please submit your revised manuscript within 30 days Jun 24 2025 11:59PM. If you will need more time than this to complete your revisions, please reply to this message or contact the journal office at plospathogens@plos.org. Please include the following items when submitting your revised manuscript:

We look forward to receiving your revised manuscript.

Kind regards,

Mary F Kearney

Academic Editor

PLOS Pathogens

Susan Ross

Section Editor

PLOS Pathogens

Sumita Bhaduri-McIntosh

Editor-in-Chief

PLOS Pathogens

orcid.org/0000-0003-2946-9497

Michael Malim

Editor-in-Chief

PLOS Pathogens

orcid.org/0000-0002-7699-2064

**Journal Requirements:**

At this stage, the following Authors/Authors require contributions: Karol Serwin, Kaja Mielczak, Anna Urbańska, Bogusz Aksak-Wąs, Malwina Karasińska-Cieślak, Piotr Ząbek, Ewa Siwak, Iwona Cielniak, Elżbieta Jabłonowska, Paweł Jakubowski, Błażej Rozpłochowski, Aleksandra Szymczak, Bartosz Szetela, Anna Kalinowska-Nowak, Monika Bociąga-Jasik, Elżbieta Mularska, Adam Witor, Anita Olczak, Władysław Łojewski, Maria Hlebowicz, and Miłosz Parczewski. Please ensure that the full contributions of each author are acknowledged in the "Add/Edit/Remove Authors" section of our submission form.

4) We are unable to open the following Supporting Information file: Supp_files_A6_dynamics.zip. Please kindly revise as necessary and re-upload.

Potential Copyright Issues:

- Figures 3, 4, and 5. Please (a) provide a direct link to the base layer of the map (i.e., the country or region border shape) and ensure this is also included in the figure legend; and (b) provide a link to the terms of use / license information for the base layer image or shapefile. We cannot publish proprietary or copyrighted maps (e.g. Google Maps, Mapquest) and the terms of use for your map base layer must be compatible with our CC BY 4.0 license.

6) Please amend your detailed Financial Disclosure statement. This is published with the article. It must therefore be completed in full sentences and contain the exact wording you wish to be published. Please ensure that the funders and grant numbers match between the Financial Disclosure field and the Funding Information tab in your submission form. Note that the funders must be provided in the same order in both places as well.

**Reviewers' Comments:**

Reviewer's Responses to Questions

**Part I - Summary**

Reviewer #1: Strengths

The study has used a large dataset of sub-subtype A6 sequences collected in Poland over a long time period.

The aim of the study is timely and address a relevant research gap

The authors employ appropriate and robust methodologies to investigate the virus’s dispersal patterns and migratory pathways.

The manuscript is well written, and the conclusions are clearly supported by the study's findings.

The study limitations are clearly stated

Weaknesses

The time calibration of the phylogenetic tree relies on a previously inferred evolutionary rate, rather than being independently estimated from the current dataset

Novelty/significance

This study makes an important contribution by providing a detailed description of HIV-1 sub-subtype A6 dispersal patterns and migration routes. It provides new insights into the molecular epidemiology of this lineage, especially regarding its spread in Poland. One central feature of this study’s importance is its timely reevaluation of how recent population movements from Ukraine, driven by the recent conflict, may have impacted both the transmission dynamics and geographic spread of the virus.

Reviewer #2: Karol Serwin et al.'s manuscript titled "Comparison of HIV-1 A6 Dispersal Dynamics in Poland Before and After the War in Ukraine" is a relevant topic underpinned with methodological rigor, making contributions to insights into the dynamics between infectious disease dynamics and geopolitical conflict. The well-integrated combination of demographic, epidemiological, and molecular data all contribute to a timely and important analysis.

Reviewer #3: Serwin et al. Parczewski present a well-described analysis of the introduction of HIV-1 sub-subtype A6 in Poland corresponding to the current ongoing war in Ukraine. It helps bring attention to the potential implications of mass population displacement at the intersection with managing infectious diseases. It is clear from the analysis, that since the Ukraine-Russian war began, there has been a significant introduction of people with HIV relocating in Poland, which has shifted the prevalence of the most common HIV-1 sub-subtype in the area. Th authors do present potential clinical implications such as long-acting CAB/RPV drug resistance. While there are specific points below there are some additional general comments that may strengthen this work.

The authors should consider including the consequences, if any of significant value, mass population displacement has on the healthcare system. While the increase of HIV-1 A6 is clear, the stress generated on healthcare workers, the systematic drug resistance genotyping, access to antiretroviral therapy are all highly important issues. While not directly related to the science presented here, it is something worth considering highlighting. Further, do the authors have data or know of other countries that are observing similar trends. If so, this would help in elevating the broader implications the war has on the epidemic.

This is highly critical work, and I thank the authors for bringing issues such as they are to the forefront against the backdrop of an ongoing war. It is vital to acknowledge and be able to understand the ripple effects and underscore the vitally important pillars of the healthcare system that help save millions of lives.

**Part II – Major Issues: Key Experiments Required for Acceptance**

Reviewer #1: No major issues

Reviewer #2: The abstract is well formatted, giving a concise summary of background, methods, key findings, and implications.

Comments:

(i) Its clarity could be enhanced by specifying the type of phylogenetic and phylogeographic analysis employed, e.g., the use of BEAST with BSSVS, in order to leave readers with a clear-cut impression of the methods.

(ii) Additionally, the term "A6 introductions" could also be explained for non-expert readers who might not be familiar with the technical terminology of HIV molecular epidemiology.

The abstract briefly communicates the overall findings of the manuscript to a broader audience and highlights the public health significance of the study. Its focus on women and the dynamics of migration is particularly relevant.

Comment: it would strengthen the account to explicitly explain why this path of analysis is so crucial in deciding intervention strategies.

Introduction makes a compelling case, placing the importance of the HIV A6 sub-subtype and its association with war-facilitated migration. The introduction effectively sets out current research and main gaps in what is currently known. Comments:

(i) to elaborate on the rationale in setting out the association between the viral A6 sub-subtype and failure to respond to antiretroviral treatment, particularly to CAB/RPV resistance.

(ii) Is there a direct association between migration patterns and therapeutic complexity that would improve cohesion?

(iii) Do the authors consider commenting on the broader implications for national health systems and surveillance efforts at the EU level?

Methods are robust, outlining a high-quality dataset and data-collection process with clear definitions. The analytical approach—phylogenetics, phylodynamics, and discrete and continuous phylogeography—is pertinent and thorough.

Comments

(i) Could the authors offer a more precise definition of cluster thresholds and definitions, either in the main text or with a clear reference to the Supplementary Materials?

(ii) Clarification is needed on whether or not temporal signal analysis (e.g., root-to-tip regression) was conducted before time-scaling.

(iii) Could the authors indicate if they expressed uncertainty in some form, e.g., as 95% HPD intervals, when comparing cluster sizes and migration events?

The Results section is dense and multi-faceted and manages to discern well between pre- and post-conflict dynamics. Figures are commendable to employ; they are educated by images and integrate well into the text.

(i) Some sections in this part are extremely crowded. Displaying key findings in concise tables will make reading easier.

(ii) Authors should include more detailed explanations of how phylogenetic "networks" were being quantified, as opposed to sequence number appearances.

(iii) In discussing singletons and dyads, describing how such groupings would be able to contribute to overall transmission network, e.g., by analyzing their potential for future growth, would be helpful.

The Discussion is logical and presents a thoughtful interpretation of the findings within the frameworks of demographic and epidemiological transitions. Reference to national data and consideration of health system factors and issues adds depth and relevance to useful public health context.

Comments and suggestions:

(i) It would be reasonable to hedge any inferred claims regarding cross-border acquisition of HIV unless stronger supporting evidence or information about Ukrainian sequence quality can be shown.

(ii) Implications for long-acting injectable therapy programs deserve more elaboration, and particularly on how resistance should be monitored.

(iii) Do the authors reflect on whether surveillance or contact tracing systems will need to be altered as a consequence of the results presented?

The Conclusion section appropriately centers on the contribution of war migration and the necessity of public health adaptation.

Comments

(i) Do the writers confirm the sentence "resources are constrained"? It might be redrafted to qualify whether this constrains capacity, finance, or both, and should preferably include suggestions for achieving maximum utilization of resources.

Reviewer #3: Lines 171-179: Can the authors comment on the rate of recombinants in Poland? While the study does focus on the influx of A6, only pol was examined, thus, breakpoints in CRFs/ URFs may be outside this area. It might be worth going into the overall HIV genetic background within Poland vs. Ukraine. While less frequent, other subtypes may be circulating and may pose an additional risk that should be considered.

Lines 209-210: In ensuring adequate mixing, I have assumed that ESS>200 was used. The Bayes factor and adjusted BF are quite appropriate as well. It would be good to include the number of iterations used in the BEAST analysis. Based on the XML it appears only 1e6 iterations were executed—what was the %burn-in?

**Part III – Minor Issues: Editorial and Data Presentation Modifications**

Reviewer #1: The time calibration of the phylogenetic tree relies on a previously inferred evolutionary rate, rather than being independently estimated from the current dataset. This approach should be explicitly acknowledged in the study’s limitations section.

Reviewer #2: Some minor improvements are also recommended.

Certain figures—particularly Figures 3 and 5—would benefit from larger font sizes and clearer legends to enhance their readability.

Lastly, the manuscript would be well-served by a final language polish from a native English speaker or professional editor to address occasional awkward phrasing and overly dense sentence constructions.

Reviewer #3: Lines 157-158: Minor point, but if within limits, it might be worth briefly explaining why the A6 sub-subtype has been identified as a potential virologic failure risk with CAB/RPV. It is great that references are included, however, some language surrounding A6 and not the main A1/A2 might be worth considering.

Lines 175-178: While the national HIV/AIDS centers for genotype A6 probably do not overlap, did the authors ensure to remove any potential identical sequences that may have been included in LANL. Being unfamiliar with Poland’s national HIV/AIDS centers, I wonder if any participants may have been part of research studies where their HIV sequence was deposited separately into LANL.

Lines 199-201, as a suggestion, please elaborate further on why large clusters were n>14 and how were the bounds of a network defined. Ref [18] is an appropriate choice, but a little more detail would be great.

In Supplemental Material, there are some minor errors (e.g., genes not italicized, extra spaces, ‘Dads’ instead of ‘Dyads’ in tables).

Supplemental Figures 2 and 3 should have labeled axes despite what they obviously state.

PLOS authors have the option to publish the peer review history of their article (what does this mean? ). If published, this will include your full peer review and any attached files.

**Do you want your identity to be public for this peer review?** For information about this choice, including consent withdrawal, please see our Privacy Policy .

Reviewer #1: No

Reviewer #2: No

Reviewer #3: No

**Figure resubmission:**
---

## [Decision Letter · Decision Letter 1]

14 Jul 2025

Dear Dr Serwin,

We are pleased to inform you that your manuscript 'Comparison of HIV-1 A6 Dispersal Dynamics in Poland Before and After the War in Ukraine' has been provisionally accepted for publication in PLOS Pathogens.

Best regards,

Mary F Kearney

Academic Editor

PLOS Pathogens

Susan Ross

Section Editor

PLOS Pathogens

Sumita Bhaduri-McIntosh

Editor-in-Chief

PLOS Pathogens

orcid.org/0000-0003-2946-9497

Michael Malim

Editor-in-Chief

PLOS Pathogens

orcid.org/0000-0002-7699-2064

Reviewer Comments (if any, and for reference):

Reviewer's Responses to Questions

**Part I - Summary**

Reviewer #1: Strengths

Large and Representative Datase

Time-Stratified Phylogeographic Approach

High-Resolution Cluster Analysis

Policy-Relevant Outcomes

Weaknesses

Assumptions in Time Calibration

Incomplete Transmission Risk Information

This study is the first large-scale phylogeographic analysis focused on war-driven HIV-1 A6 spread into an EU country. It Identified distinct epidemiological shifts attributable to a real-time humanitarian crisis.

Reviewer #2: (No Response)

Reviewer #3: No comments.

**Part II – Major Issues: Key Experiments Required for Acceptance**

Reviewer #1: Not applicable

Reviewer #2: (No Response)

Reviewer #3: No comments.

**Part III – Minor Issues: Editorial and Data Presentation Modifications**

Reviewer #1: Not applicable

Reviewer #2: (No Response)

Reviewer #3: No comments.

PLOS authors have the option to publish the peer review history of their article (what does this mean? ). If published, this will include your full peer review and any attached files.

**Do you want your identity to be public for this peer review?** For information about this choice, including consent withdrawal, please see our Privacy Policy .

Reviewer #1: No

Reviewer #2: **Yes: ** Jorge Quarleri

Reviewer #3: No

---

## [Editor Report · Acceptance letter]

Dear Dr Serwin,

We are delighted to inform you that your manuscript, " 

Comparison of HIV-1 A6 Dispersal Dynamics in Poland Before and After the War in Ukraine," has been formally accepted for publication in PLOS Pathogens.

Best regards,

Sumita Bhaduri-McIntosh

Editor-in-Chief

PLOS Pathogens

orcid.org/0000-0003-2946-9497

Michael Malim

Editor-in-Chief

PLOS Pathogens

orcid.org/0000-0002-7699-2064